# A Retrospective Observational Study of Pulmonary Impairments in Long COVID Patients

**DOI:** 10.3390/biomedicines14010145

**Published:** 2026-01-10

**Authors:** Lanre Peter Daodu, Yogini Raste, Judith E. Allgrove, Francesca I. F. Arrigoni, Reem Kayyali

**Affiliations:** 1Faculty of Health, Science, Social Care and Education, Kingston University, London KT1 2EE, UK; f.arrigoni@kingston.ac.uk (F.I.F.A.); r.kayyali@kingston.ac.uk (R.K.); 2Chest/Respiratory Department, Croydon University Hospital, Croydon Health Services NHS Trust, London CR7 7YE, UK; yoginiraste@nhs.net; 3School of Allied Health and Exercise Sciences, Faculty of Health, Environment and Medical Sciences, Bournemouth University, Bournemouth BH12 5BB, UK; jallgrove@bournemouth.ac.uk

**Keywords:** long COVID, pulmonary function, diffusion impairment, ground-glass opacities, restrictive pattern, chest CT scan, alveolar gas exchange

## Abstract

**Background/Objective:** Pulmonary impairments have been identified as some of the most complex and debilitating post-acute sequelae of SARS-CoV-2 infection (PASC) or long COVID. This study identified and characterised the specific forms of pulmonary impairments detected using pulmonary function tests (PFT), chest X-rays (CXR), and computed tomography (CT) scans in patients with long COVID symptoms. **Methods:** We conducted a single-centre retrospective study to evaluate 60 patients with long COVID who underwent PFT, CXR, and CT scans. Pulmonary function in long COVID patients was assessed using defined thresholds for key test parameters, enabling categorisation into normal, restrictive, obstructive, and mixed lung-function patterns. We applied exact binomial (Clopper–Pearson) 95% confidence intervals to calculate the proportions of patients falling below the defined thresholds. We also assessed the relationships among spirometric indices, lung volumes, and diffusion capacity (DLCO) using scatter plots and corresponding linear regressions. The findings from the CXRs and CT scans were categorised, and their prevalence was calculated. **Results:** A total of 60 patients with long COVID symptoms (mean age 60 ± 13 years; 57% female) were evaluated. The cohort was ethnically diverse and predominantly non-smokers, with a mean BMI of 32.4 ± 6.3 kg/m^2^. PFT revealed that most patients had preserved spirometry, with mean Forced Expiratory Volume in 1 Second (FEV1) and Forced Vital Capacity (FVC) above 90% predicted. However, a significant proportion exhibited reductions in lung volumes, with total lung capacity (TLC) decreasing in 35%, and diffusion capacity (DLCO/TLCO) decreasing in 75%. Lung function pattern analysis showed 88% of patients had normal function, while 12% displayed a restrictive pattern; no obstructive or mixed patterns were observed. Radiographic assessment revealed that 58% of chest X-rays were normal, whereas CT scans showed ground-glass opacities (GGO) in 65% of patients and fibrotic changes in 55%, along with findings such as atelectasis, air trapping, and bronchial wall thickening. **Conclusions:** Spirometry alone is insufficient to detect impairment of gas exchange or underlying histopathological changes in patients with long COVID. Our findings show that, despite normal spirometry results, many patients exhibit significant diffusion impairment, fibrotic alterations, and ground-glass opacities, indicating persistent lung and microvascular damage. These results underscore the importance of comprehensive assessment using multiple diagnostic tools to identify and manage chronic pulmonary dysfunction in long COVID.

## 1. Introduction

Pulmonary impairments have been identified as some of the most complex and debilitating post-acute sequelae of SARS-CoV-2 infection (PASC) or long COVID. The incidence of long COVID is significant; findings suggest that 10% to 70% of those who have recovered from COVID-19 continue to suffer from new or existing symptoms [1,2,3]. The disease presents a broad spectrum of incapacitating symptoms that affect several bodily systems, including the respiratory, cardiovascular, neurological, gastrointestinal, and musculoskeletal systems [4,5,6]. The intricate nature of long COVID is evident in its association with more than 200 symptoms, requiring more investigation into its effects on the human body [7,8,9]. Survivors of severe COVID-19, especially those who suffered from acute respiratory distress syndrome (ARDS), often develop long-term pulmonary complications such as pulmonary fibrosis, which arises from excessive collagen deposition in the extracellular matrix of the lung parenchyma [10,11]. The mechanical stress imposed by invasive ventilation during ICU stays further exacerbates lung injury, contributing to the development of chronic pulmonary conditions [12,13]. This aligns with observations that patients recovering from ARDS frequently exhibit impaired lung function, which can persist for months or even years post-infection [14]. Continued dyspnoea and fatigue are concerning, particularly in patients who initially had abnormal lung function [15].

In addition to the wide array of symptoms and multisystem involvement, it is important to acknowledge the complexity of differential diagnosis in patients presenting with symptoms following acute SARS-CoV-2 infection [16,17]. Symptoms that emerge or persist for weeks or months after the initial infection may not be exclusively attributable to long COVID but could also indicate the presence or exacerbation of other medical conditions, such as pre-existing chronic respiratory diseases, cardiovascular complications, post-intensive care syndrome, or unrelated new-onset illnesses. This diagnostic complexity highlights the need for a thorough clinical assessment and judicious use of diagnostic investigations to differentiate long COVID from alternative or co-existing diagnoses, ensuring that patients receive appropriate and targeted management.

Also, the ongoing challenge of classifying post-COVID syndromes, particularly long COVID, lies in the breadth and complexity of symptoms that span multiple organ systems and the lack of consensus regarding their definitions and diagnostic criteria [18,19,20]. While numerous studies have documented the persistent and often debilitating symptoms following SARS-CoV-2 infection, the heterogeneity of patient presentations and overlap with other chronic conditions have made it difficult to establish clear, universally accepted classifications. This uncertainty is compounded by the variable use and interpretation of diagnostic tools, such as PFTs, CXRs, and CT scans, which can lead to under-recognition or mischaracterisation of post-COVID pulmonary impairments [21,22,23].

A significant percentage of patients experiencing long COVID have demonstrated a low diffusing capacity of the lungs for carbon monoxide (DLCO) alongside normal chest scans, possibly stemming from pulmonary endothelial dysfunction and microthrombosis [24,25]. Individuals with COVID-19 experience improvements in lung function over time, with significant recovery in DLCO and other measures of pulmonary function observed between three and seventeen months following infection [26,27]. CT scans of COVID-19 survivors reveal persistent inflammation in up to 71% of cases two to three months after hospital discharge [28]. These signs of persistent inflammation include bronchial wall thickening and areas of air trapping. These inflammatory features suggest ongoing pulmonary involvement and may contribute to the chronic symptoms observed in patients with long COVID. A systematic review showed that patients with long COVID frequently exhibit interstitial fibrosis, pleural abnormalities, and airway changes, which can be classified into distinct imaging categories [29]. Three to six months after infection, a significant number of patients continued to exhibit abnormal CT imaging findings [27]. The lasting effects of COVID-19 on the respiratory system can be substantial, often manifesting as impaired lung function and persistent symptoms such as dyspnoea and fatigue.

Despite the growing number of studies on long COVID, a critical gap remains in our understanding of its long-term respiratory consequences, the mechanisms underlying these impairments, and the most effective strategies for assessing and managing them. Also, studies have highlighted the persistence of breathlessness, reduced exercise capacity, and abnormal pulmonary function in individuals who have ostensibly recovered from the acute infection. However, heterogeneity in patient populations, inconsistencies in diagnostic criteria, and variations in follow-up periods have led to a fragmented evidence base. Also, studies focus on single diagnostic modalities or lack detailed characterisation of lung function and structural abnormalities, and studies that focus on pulmonary impairments using a combination of PFTs, CXRs, and CT scans are limited [22,23]. This contributes to a fragmented understanding of the respiratory sequelae of long COVID. This has hindered the development of robust clinical guidelines, leaving clinicians uncertain when treating affected individuals. Understanding the prevalence and nature of these impairments is essential for monitoring and managing patient care.

This study identified and characterised the specific forms of pulmonary impairments detected using PFTs, CXRs, and CT scans in patients with long COVID symptoms. These multimodal approaches (PFTs, CXRs, and CT scans) helped to identify specific patterns and severities of impairments that might be missed if only one diagnostic method were used.

## 2. Materials and Methods

### 2.1. Research Design

This was a single-centre, retrospective study of patients managed for long COVID symptoms between April 2021 and December 2023 at a University Hospital in London.

The inclusion criteria included adult patients (≥18 years old) who were managed for long COVID and underwent PFTs with/without CXR and CT scans. The study excluded all other long COVID patients without PFTs. The consultant physicians confirmed the diagnosis of long COVID and recommended PFT, CXR, and CT scans based on the patient’s clinical condition. The diagnosis of long COVID was established based on current clinical criteria, including the persistence or emergence of symptoms attributable to SARS-CoV-2 infection beyond 12 weeks from the initial onset, without an alternative explanation. Patients included in the study were diagnosed at least 12 weeks after their acute infection. The study was confined to the use of existing, de-identified data.

### 2.2. Pulmonary Function Test Assessment

The thresholds for PFT parameters were defined to identify reduced values. Specifically, thresholds were set at 80% for Forced Expiratory Volume in 1 Second (FEV1), Forced Vital Capacity (FVC), Total Lung Capacity (TLC), Diffusing Capacity of the Lungs for Carbon Monoxide—Single Breath (TLCO_SB), and Vital Capacity (VC), and at 75% for the ratio of Forced Expiratory Volume in 1 Second to Vital Capacity (FEV1/VC) [21]. Using these thresholds, the percentage of patients was calculated for values below each threshold. These percentages represented the proportion of patients whose lung function parameters were below the defined thresholds.

Patients were classified into lung-function pattern categories (normal, restrictive, obstructive, and combined) using prespecified spirometric and gas-transfer thresholds [30]. Pulmonary function patterns were defined as shown in Table 1. Cases that did not meet any of these criteria were labelled “normal”. The classification algorithm required non-missing values for the component tests used in each rule; rows with missing component values were not assigned to a category unless the remaining criteria could be evaluated. The precedence of assignment was Restrictive first, then Obstructive, Combined, and finally Normal, to avoid overlapping assignments.

### 2.3. Data Source and Management

The data source was the electronic patient record (EPR). We collected patients’ demographic information, spirometry: % Predictive Value for FEV1, FVC, VC, FEV1/VC, Peak Expiratory Flow (PEF), Maximal Mid-Expiratory Flow (MMEF75/25), lung Volume: TLC, expiratory Reserve Volume (ERV), residual volume (RV), RV/TLC, diffusion Capacity: % Predictive Value for TLCO_SB, alveolar volume (AV) single Breath and Imaging: Chest X-ray and CT scan findings. Data were extracted and managed using Castor Electronic Data Capture (EDC).

### 2.4. Study Outcomes

The primary outcome was to identify and characterise the specific forms of pulmonary impairments in patients with long COVID symptoms using PFT. The secondary outcomes were to identify and characterise specific forms of pulmonary impairments in patients with long COVID symptoms using CXR and CT scans.

### 2.5. Ethical Approval

The Health Research Authority (HRA) in England and Health and Care Research Wales (HCRW) granted ethical approval for the research study under the Research Ethics Committee reference number 23/HRA/1637.

### 2.6. Statistical Analysis

The data was analysed using R version 4.3.3. Multiple imputation by chained equations (MICE) was employed to address missing data in this study. Five imputed datasets were generated using the predictive mean matching (PMM) method, with 50 iterations and a seed value of 500 to ensure reproducibility.

Descriptive statistics were used to summarise the dataset’s key characteristics. Continuous variables were summarised using their mean and standard deviation, while categorical variables were presented with their counts and percentages. The resulting descriptive statistics facilitated a deeper understanding of the patient population and set the stage for further statistical analyses.

Counts for each pulmonary function pattern were obtained from the labelled data and converted to proportions. Percentages were calculated as (count/N) × 100, where N is the total number of patients with available pattern classification (N = 60). Exact binomial (Clopper–Pearson) 95% confidence intervals for the proportions were calculated because some cell counts were small; the Clopper–Pearson method provides exact coverage for proportions in small samples and is preferable to normal approximations when observed events are few. Exact confidence intervals and binomial tests were computed in R using the base function binom.test. Sensitivity analyses were performed by varying the FEV1/FVC threshold used to define obstruction (70% versus 75%) and by repeating the classification after removing the requirement for reduced TLCO in the obstructive definition. Pattern distributions across scenarios were compared descriptively. The relationships between spirometry values, lung volumes, and diffusion capacity parameters were assessed using the Spearman correlation coefficient.

The clinical characteristics of long COVID patients’ CXR and CT scan findings were systematically evaluated. The radiological and tomographical observations were categorised into key groups that reflected common patterns, such as ground-glass opacities, fibrosis, or airway changes. For each category, both the number and percentage of patients exhibiting these features were calculated, providing insight into the frequency with which each type of abnormality occurred in the study population. This approach enabled a comprehensive visual and quantitative assessment of lung involvement, complementing pulmonary function tests and clarifying the range and severity of respiratory manifestations associated with long COVID. The findings were reported in accordance with the standards outlined in the Strengthening the Reporting of Observational Studies in Epidemiology (STROBE) guidelines.

## 3. Results

### 3.1. Characteristics of the Study Participants

A total of 60 patients with long COVID symptoms were included in the study. The mean age of the patients was 60 ± 13 years (Table 2); 57% female and 43% male. The mean BMI was 32.4 ± 6.3 kg/m2. Forty per cent were White, 27% Asian, 20% Black and 13% were of any other ethnicity. Most patients (67%) were non-smokers, 30% were ex-smokers, and 3.3% currently smoke. 35% of patients were on different medications.

The spirometry means measures: FEV1 was 92% (±21%), FVC was 94% (±23%), and VC was 92% (±22%), the ratio of FEV1 to VC (FEV/VC) was 102% (±14%), PEF was 100% (23%), MMEF75/25 was 93% (±44%) of the predicted values. The lung volume means: TLC was 86% (±18%), the ERV was 95% (±52%), the RV was 83% (±25%), and the ratio of RV to TLC (RV/TLC) was 98% (±19%) of the predicted values. The diffusion capacity means: TLCO SB was 83% (±92%), and the VA Single Breath was 77% (±17%) of the predicted values.

### 3.2. Pulmonary Function Test

Table 3 shows a decrease in FEV1 among 30% of patients. FVC abnormalities were observed in 25% of patients (*p* < 0.001). Among the patients, a reduction in the FEV/VC ratio was observed in 3.33%. TLC was significantly decreased (*p* < 0.001) in 35% of the patient population. TLCO showed the most significant decrease, affecting 75% of patients, while VC decreased by 25%.

### 3.3. Lung Function Patterns

Figure 1 shows that most individuals with long COVID had “normal” lung function, while a small minority had a “restrictive pattern”. 53 out of 60 patients were labelled Normal, which is 88.33% (95% Clopper–Pearson confidence interval 77.43% to 95.18%). Seven out of 60 patients were labelled Restrictive, which is 11.67% (95% Clopper–Pearson confidence interval 4.82% to 22.57%). There were no cases labelled Obstructive or combined/mixed patterns. Sensitivity analyses that adjusted the FEV1/FVC cut-off from 70% to 75% did not alter the distribution of lung-function patterns (Table 4), with 53 patients classified as normal and 7 as restrictive under both thresholds. However, removing the requirement for reduced TLCO in the obstructive definition resulted in one additional patient being classified as obstructive, while the number of restrictive cases remained unchanged. The obstructive physiology was uncommon and was observed in only one patient, even under the more permissive classification criteria.

Figure 2 displays the relationships between various spirometric measurements using five scatter plots with corresponding linear regression lines. Plot (a) shows the relationship between PEF (% Predicted) and FVC (% Predicted). The data points were distributed around a positively sloped regression line, suggesting a positive correlation between PEF and FVC (i.e., as FVC increases, PEF tends to increase as well); however, this correlation was not statistically significant (*p* > 0.05). Plot (b) illustrates the relationship between FEV_1_ (% Predicted) and FVC (% Predicted). The scatter plot showed a positive correlation, with data points clustered around an upward-sloping regression line, indicating that higher FVC values are associated with higher FEV_1_ values. This correlation was statistically significant (*p* < 0.05). Plot (c) demonstrates the relationship between FVC (% Predicted) and TLC (% Predicted). The data points were scattered around a positively sloped regression line, suggesting a positive correlation between FVC and TLC, indicating that FVC also tends to increase as TLC increases. This correlation was statistically significant (*p* < 0.05). Plot (d) shows the relationship between RV (% Predicted) and TLC (% Predicted). The scatter plot indicated a positive correlation, with data points distributed around an upward-sloping regression line, suggesting that higher TLC values are associated with higher RV values. This correlation was statistically significant (*p* < 0.05). Plot (e) depicts the relationship between the RV/TLC Ratio (% Predicted) and TLC (% Predicted). The data points were scattered around a negatively sloped regression line, indicating a negative correlation between the RV/TLC Ratio and TLC, with the RV/TLC ratio decreasing as TLC increases. This negative correlation was statistically significant (*p* < 0.05).

### 3.4. Radiographic and Tomographic Findings

CXR findings (Figure 3) showed that 58.3% of patients had clear lung fields, while 11.7% had interstitial changes, atelectasis (8.3%), fibrotic changes (6.7%), opacities/infiltrates (5.0%), and other changes (3.3%).

Tomographic findings (Figure 4) revealed that only 22.5% had no significant CT findings. Most (65%) patients’ CT scans showed ground-glass opacities (GGOs), with 30% showing widespread involvement, 20% localised, and 15% residual. Fibrotic changes were observed in 55% of cases, including interstitial changes (25%), localised fibrosis (17.5%), and diffuse fibrosis (12.5%). In addition, 32.5% of patients displayed atelectasis and air trapping, including 15% with patchy atelectasis, 10% with curvilinear atelectasis, and 7.5% with mosaic attenuation. Nodules and nodularity were found in 12.5% of cases (7.5% perilymphatic and 5.0% pulmonary). Bronchial wall thickening was observed in 10%, pleural effusions in 5%, and post-infective changes in 7.5% of the patients.

## 4. Discussion

The present study characterised the chronic pulmonary phenotype in a cohort of patients experiencing long COVID, or Post-Acute Sequelae of SARS-CoV-2 Infection (PASC). The core findings reveal a critical dissociation: preserved bulk airflow mechanics co-exist with profound impairment in alveolar gas exchange, suggesting that persistent parenchymal and microvascular damage is the dominant pathological signature in this patient group. The most significant physiological abnormality observed was the severely reduced diffusing capacity for carbon monoxide (TLCO), found in 75% of the cohort. This high prevalence is consistent with the higher range of reported rates in post-COVID survivors [31] and identifies severe diffusion impairment as the key physiological feature. In stark contrast, mean spirometry values were largely normal (${\mathrm{FEV}}_{1}$ at 92% predicted, FVC at 94% predicted), and only 3% of patients exhibited a decreased ${\mathrm{FEV}}_{1}/\mathrm{VC}$ ratio, effectively minimising generalised, clinically significant obstructive disease as the primary cause of chronic respiratory morbidity [32]. This dissociation emphasises that TLCO, which assesses the functional integrity of the alveolar–capillary unit, is an essential biomarker for identifying significant underlying parenchymal and microvascular injury in PASC patients, where routine spirometry may provide false reassurance [33].

While diffusion impairment was dominant, intrinsic restrictive lung disease was also observed, with 35% of patients showing reduced Total Lung Capacity (TLC) and 12% meeting the criteria for a formal restrictive ventilatory pattern. This finding aligns with the pulmonary sequelae reported following previous coronavirus epidemics (SARS-CoV-1 and MERS), where decreases in TLC (5.2–10.9% of cases) and TLCO (15.5–43.6% of cases) were documented months to years after acute infection [34]. The co-occurrence of reduced TLC and severe TLCO reduction establishes a “diffusion-impaired restriction” phenotype in a subset of the cohort, often linked to severe acute infection courses [35]. Furthermore, while lung function parameters improved in longitudinal studies, especially in those with severe disease [32], the persistence of deficits in this chronic-phase cohort suggests stable or potentially irreversible damage.

The functional abnormalities observed are strongly correlated with persistent structural damage identified on Computed Tomography (CT) scans, where only 23% of patients had no significant findings. The most frequent abnormality was persistent ground-glass opacities (GGOs), observed in 65% of patients. While GGOs can reflect residual inflammation or an Organising Pneumonia (OP) pattern that may resolve [36], their high prevalence in the chronic phase, combined with the presence of fibrotic changes in 55% of patients, confirms a high burden of Post-COVID Interstitial Lung Disease (PC-ILD). Fibrotic changes, including interstitial, localised, and diffuse fibrosis, represent irreversible architectural distortion [37] and are associated with advanced age and disease severity [38]. The reduction in TLCO shows explicitly a high specificity (approximately 88%) for predicting these fibrotic-like changes on CT scans.

The high prevalence of fibrotic changes (55%) versus formal restriction (12%) may be partly reconciled by the concurrent finding of air trapping in 33% of cases. Air trapping is a radiographic marker of Small Airway Dysfunction (SAD) [31]. SAD leads to non-uniform gas distribution and hyperinflation, which results in increased Residual Volume (RV) [39]. This increased RV can artificially elevate the measured TLC, potentially masking volume loss due to parenchymal scarring [40]. The presence of SAD, often undetected by routine spirometry [39], is a significant contributor to dyspnea and TLCO reduction via ventilation–perfusion (V/Q) mismatch. Sensitive techniques like impulse oscillometry (IOS) or multiple breath washout (MBW) are needed to objectively quantify SAD [41].

Mechanistically, the severe reduction in TLCO primarily points to injury of the alveolar–capillary membrane or the pulmonary capillary volume (Pcap) [35]. Post-COVID pathology is known to involve diffuse endothelial injury and persistent pulmonary microthrombosis, driven by factors such as ongoing excessive Neutrophil Extracellular Trap (NET) formation [42]. The reduction in effective Pcap drastically reduces TLCO. While the absence of the transfer coefficient (KCO or TLCO/VA) is a limitation, its inclusion in future analyses is crucial, as a reduced TLCO with a preserved KCO suggests reduced alveolar volume or perfusion defects, which may be reversible, whereas reduced TLCO alongside a reduced KCO is highly suggestive of irreversible fibrosis [38]. Based on the 55% fibrosis rate, a portion of this cohort likely exhibits the latter, irreversible pattern [42].

The demographic profile of the cohort—average age 60 years and a high mean Body Mass Index (BMI) of $32.4{\;\mathrm{kg}/m}^{2}$—is relevant. Obesity is an independent risk factor for severe acute COVID-19 and subsequent PASC development [43], and may worsen outcomes due to its pro-inflammatory and pro-thrombotic state [44]. Moreover, obesity independently imposes an extrinsic restrictive mechanical load, which can contribute to reduced TLC and FVC [45], thereby confounding the distinction between intrinsic fibrotic restriction and extrinsic mechanical restriction. The older average age is also consistent with established findings linking increased age to more persistent structural changes, including GGOs and fibrotic changes, up to two years post-infection [45]. Furthermore, the ethnic diversity (20% Black) necessitates awareness of documented health disparities, as non-Hispanic Black patients have been associated with a statistically significant lower DLCO% predicted compared to White and Hispanic patients in post-COVID cohorts, potentially compounded by lower access to essential care like pulmonary rehabilitation [46]. Also, studies have documented that non-Hispanic Black patients tend to have a statistically significant lower DLCO per cent predicted compared to White and Hispanic patients following COVID-19 infection [47]. Several factors may contribute to this disparity, including genetic predispositions, differences in baseline lung function, and higher prevalence of comorbidities that affect lung health [48]. Measurement and interpretation issues (for example, race-based adjustments in spirometry reference equations) may also contribute to under-recognition or misclassification of impairment in some groups [49].

The relationship between the actual severity of COVID-19 infection and subsequent pulmonary function is a critical aspect of understanding long COVID and its long-term respiratory consequences. Previous research indicates that individuals who suffered from more severe forms of COVID-19, particularly those requiring hospitalisation or intensive care, often exhibit greater degrees of pulmonary impairment—such as reduced diffusing capacity (DLCO), restrictive patterns, and evidence of fibrosis—compared to those with milder disease presentations [50,51,52]. This is likely attributable to the extent of acute lung injury, prolonged hypoxemia, and the effects of mechanical ventilation, all of which may contribute to long-term structural and functional changes in the lungs.

These findings translate into clear clinical recommendations. Comprehensive pulmonary function testing, including TLC and TLCO, must be mandatory for symptomatic PASC patients. Given the high rate of structural abnormalities, CT imaging is strongly indicated for patients with persistent or worsening respiratory symptoms lasting for more than 3 months [53,54]. The presence of 65% GGOs (suggestive of organising pneumonia) and 55% fibrotic changes mandates careful therapeutic consideration. Also, Pulmonary Rehabilitation (PR) remains a crucial cornerstone, providing functional gains in DLCO and exercise capacity, especially valuable for this older, high-BMI cohort with persistent functional limitations.

The limitations of this study include the lack of pre-COVID PFT data and detailed longitudinal follow-up, which prevents definitive assessment of progression or the exclusion of pre-existing conditions. The absence of a carbon monoxide transfer coefficient and data restricts the precise mechanistic characterisation of the DLCO deficit, and reliance on conventional spirometry may underestimate the true prevalence of SAD. Without baseline or pre-COVID pulmonary assessments, it is difficult to determine the extent to which observed impairments are directly attributable to SARS-CoV-2 infection or represent pre-existing conditions. Although a comprehensive set of pulmonary function tests and imaging studies was performed, the study did not include advanced techniques such as hyperpolarised gas MRI or impulse oscillometry, which may offer greater sensitivity in detecting small airway disease.

## 5. Conclusions

In conclusion, this study demonstrates that long COVID is characterised by a striking dissociation between preserved spirometric measures and severe impairment in alveolar gas exchange, with a high prevalence of diffusion deficits and persistent radiological abnormalities. The dominance of reduced TLCO and the frequency of fibrotic and ground-glass changes on CT scans highlight ongoing parenchymal and microvascular injury as central features of post-acute sequelae. Notably, intrinsic restrictive patterns and small airway dysfunction contribute further to the complex pulmonary phenotype observed. The demographic profile, older age, elevated BMI, and ethnic diversity reflect known risk factors for both acute severity and chronicity of symptoms. These findings highlight the necessity of comprehensive pulmonary function testing and routine CT imaging in symptomatic long COVID patients, alongside targeted therapeutic interventions such as pulmonary rehabilitation. Limitations, including the lack of pre-COVID baseline data and advanced small airway assessment, warrant further longitudinal research to clarify the trajectory and reversibility of these sequelae. Continuous surveillance and multidisciplinary management remain essential to address the significant and potentially enduring burden of chronic pulmonary dysfunction in this population.

## Figures and Tables

**Figure 1 biomedicines-14-00145-f001:**
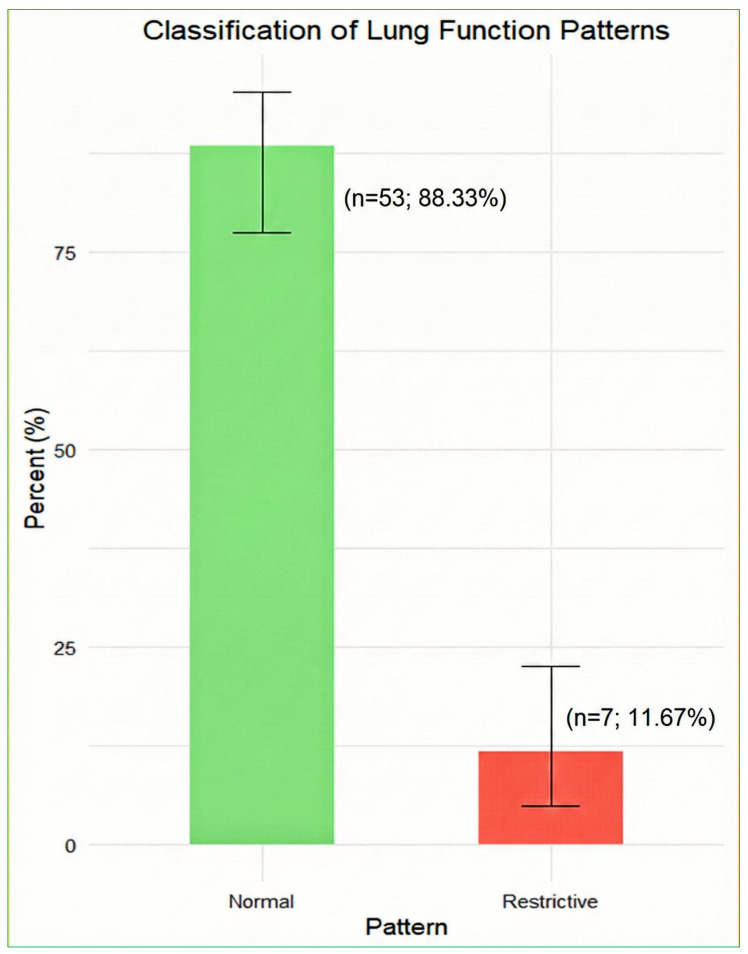
Proportion of patients with normal and restrictive lung-function patterns (*N* = 60). Percentages are shown with exact (Clopper–Pearson) 95% confidence intervals.

**Figure 2 biomedicines-14-00145-f002:**
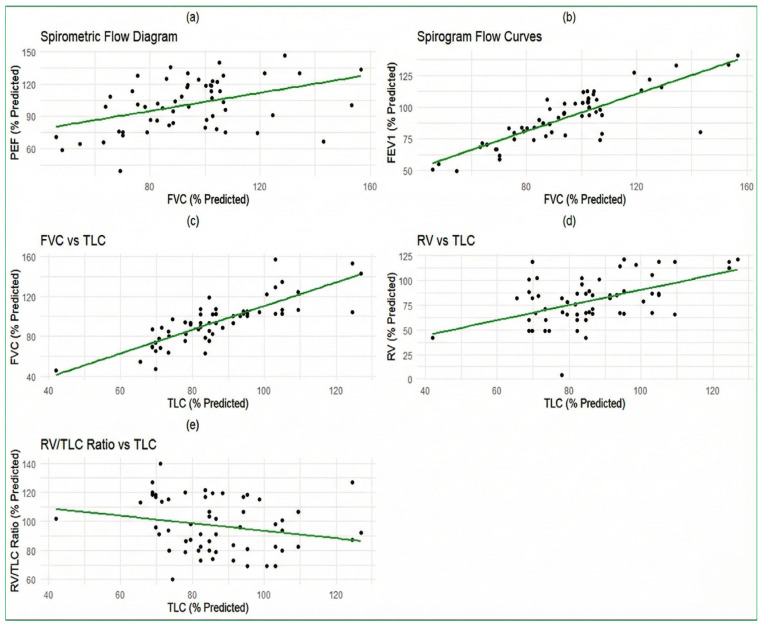
(**a**) spirometric flow diagram showing a relationship between PEF and FVC; (**b**) spirogram flow curves illustrate the relationship between FEV1 and FVC; (**c**) the relationship between FVC and TLC; (**d**) the relationship between RV and TLC; (**e**) the relationship between RV/TLC ratio and TLC. The black dots represent individual patient data points. The green line represents the linear regression line of best fit, demonstrating the correlation between the variables in each panel.

**Figure 3 biomedicines-14-00145-f003:**
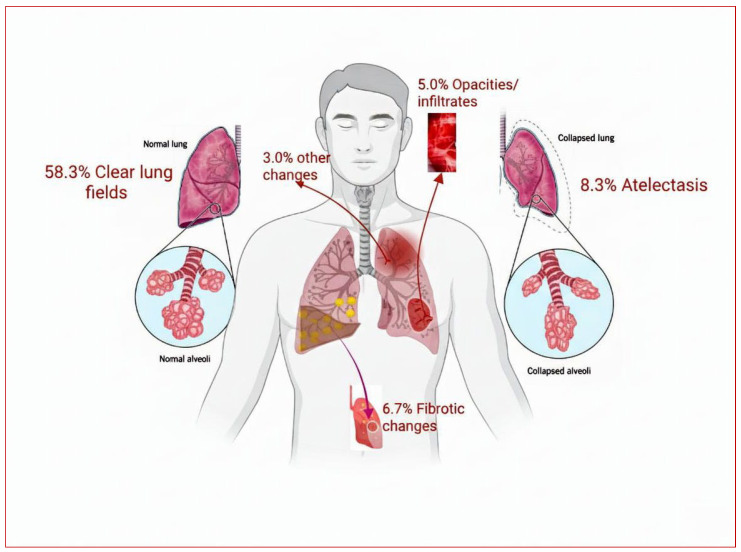
Chest X-ray of Long COVID Patients showing pulmonary abnormalities. Percentages denote the proportion of patients with each chest imaging finding. Clear lung fields indicate no radiological abnormalities. Opacities/infiltrates represent areas of increased attenuation consistent with inflammatory changes. Atelectasis indicates partial or complete lung collapse, while fibrotic changes reflect imaging features suggestive of pulmonary fibrosis. Images are schematic and not to scale.

**Figure 4 biomedicines-14-00145-f004:**
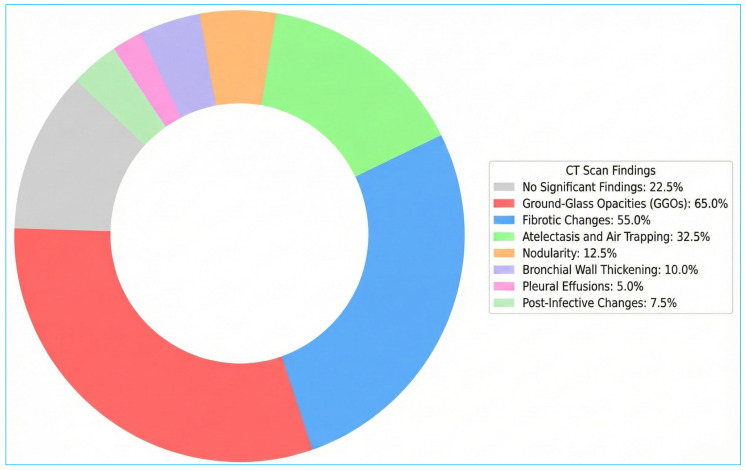
Chest CT Scan Findings in Long COVID Patients. GGOs were seen in most CT scans: 30% widespread, 20% localised, 15% residual. Fibrosis occurred in 55% of cases: interstitial (25%), localised (17.5%), and diffuse (12.5%).

**Table 1 biomedicines-14-00145-t001:** Criteria for classification of pulmonary function test patterns.

Pulmonary Function Pattern	Diagnostic Criteria
Restrictive pattern	FVC < 80% predicted; TLC < 80% predicted; TLCO (DLCO single-breath) < 80% predicted; preserved FEV1 (≥65% predicted); FEV1/FVC ratio ≥ 70%
Obstructive pattern	FEV1 < 80% predicted; FEV1/FVC ratio < 70%; preserved lung volumes (TLC ≥ 80% predicted or FVC ≥ 80% predicted); TLCO < 80% predicted
Combined pattern	Obstructive physiology (FEV1 < 80% predicted and FEV1/FVC ratio < 70%) with reduced lung volumes (FVC < 80% predicted or TLC < 80% predicted)

FVC (forced vital capacity); FEV1 (forced expiratory volume in one second); TLC (total lung capacity); TLCO (DLCO) (transfer factor of the lung for carbon monoxide (single-breath method)). Diagnostic thresholds were defined according to established pulmonary function test guidelines [30].

**Table 2 biomedicines-14-00145-t002:** Characteristics of the study of long COVID patients.

Variables	*N* = 60
Age (years) ^1^	60(13)
Gender ^2^	
Female	34 (57%)
Male	26 (43%)
BMI (kg/m^2^) ^1^	32.4 (6.3)
Ethnicity ^2^	
White	24 (40%)
Black	12 (20%)
Asian	16 (27%)
Any other ethnic	8 (13%)
Smoking status ^2^	
Non-smoker	40 (67%)
Ex-smoker	18 (30%)
Smoker	2 (3.3%)
Patient on medications ^2^	21(35%)
PFT	
Spirometry ^1^	
FEV1	92 (21)
FVC	94 (23)
VC	92 (22)
FEV/VC	102 (14)
PEF	100 (23)
MMEF75/25	93 (44)
Lung volume ^1^	
TLC	86 (18)
ERV	95 (52)
RV	83 (25)
RV/TLC	98 (19)
Diffusion capacity ^1^	
TLCO SB	83 (92)
VA Single Breath	77 (17)
^1^ Mean (SD); ^2^ n (%)	

Values are presented as: ^1^ = Mean (Standard Deviation); ^2^ = Number (Percentage). Spirometry, lung volumes, and diffusion capacity results are expressed as percentages of predicted values.

**Table 3 biomedicines-14-00145-t003:** Percentage of patients with reduced pulmonary function parameters.

Parameter	Percentage of Patients	*p*-Value *
FEV1	30.00%	*p* < 0.001
FVC	25.00%	*p* < 0.001
FEV1/VC	3.33%	*p* < 0.001
TLC	35.00%	*p* < 0.001
TLCO	75.00%	*p* < 0.001
VC	25.00%	*p* < 0.001

***** Fisher’s exact test. *p*-values indicate statistical significance of pulmonary function parameter abnormalities. Percentages represent the proportion of patients with abnormal test results. FEV1 (forced expiratory volume in one second); FVC (forced vital capacity); FEV1/VC (ratio of forced expiratory volume in one second to vital capacity); TLC (total lung capacity); TLCO (transfer factor of the lung for carbon monoxide).

**Table 4 biomedicines-14-00145-t004:** Sensitivity analysis of lung-function pattern classification under alternative definitions.

Sensitivity Scenario	Normal, n (%)	Restrictive, n (%)	Obstructive, n (%)
FEV1/FVC < 70%, TLCO required	53 (88.3)	7 (11.7)	0 (0.0)
FEV1/FVC < 75%, TLCO required	53 (88.3)	7 (11.7)	0 (0.0)
FEV1/FVC < 70%, TLCO not required	52 (86.7)	7 (11.7)	1 (1.7)

FEV_1_ (forced expiratory volume in 1 s); FVC (forced vital capacity); TLCO (transfer factor of the lung for carbon monoxide). Percentages were calculated using a total sample size of n = 60.

## Data Availability

The raw data supporting the conclusions of this article will be made available by the authors on request.

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
