# Peer review of "A Retrospective Observational Study of Pulmonary Impairments in Long COVID Patients"

_biomedicines, 2026, doi:10.3390/biomedicines14010145_

Round 1

Reviewer 1 Report

Comments and Suggestions for Authors

Congratulations to authors , a very interesting and original protocol including all aspects of lung damages occurring in covid and post covid infections .

just two personal questions :

why don’t they to choose pulmonary scintigraphy to detect symptomatic or asymptomatic associated pulmonary embolism?

how did they suggest to approach to respiratory rehab in outpatients ? 

Reviewer 2 Report

Comments and Suggestions for Authors

Dear authors, please find the comments I considered to be useful, for this study that could have been also useful, if several problems will be adressed, carefully.

The title” The title is common, I propose to rewrite the title, more dynamic and more applied to the content. There are hundreds of studies on the subject, and the current title does not make it stand out at all.

The abstract: The correctly accepted and used terminology is Long Covid, not long Covid.

In the conclusions, the writing style is not adapted to the content. Rephrase the conclusions, being precise and concise in what you want to express. Example: Spirometry does not help identify the degree of deterioration of gas exchange, nor in highlighting histopathological alterations.

Key words: 5-7, no more. I would drop "fibrotic changes", it is too vague and without reference to the lung.

Introduction: it needs to be redone, summarizing what is important in the problem of aboradta-classifications of post-Covid syndromes; what is not yet clear and what this study brings new, so that we have reason to read it with interest.

Material and method: 60 patients explored only by spirometry, with random imaging exploration, which increases the percentage of error in statistical interpretation and decreases the value of the study.

Please answer the following questions, which are substantive issues of scientific content: what kind of doctors diagnosed Long Covid? What specialty and training do they have in this new pathology? What are the diagnostic criteria used/at what exact period of their evolution were they diagnosed, how long after infection?

2.2.Data source: in fact, you have used as a paraclinical diagnostic method mainly pulmonary functional explorations and gas distribution. Why did you not use the NO fraction in exhaled air, it is a biomarker that strongly correlates with respiratory dysfunction and the evolution towards asthma, for allergic patients? How many patients were allergic out of the 60? It is known that the evolution of Covid in allergic patients was different.

There is no homogeneity in the functional diagnostic criteria, because not all patients initially had imaging exploration; why didn't you use MRI, is it optimal for histopathological changes in COVID?

Lines 136-157 do not belong to the statistical analysis section. Their insertion is erroneous, proving that the authors did not pay enough attention to the content and the supervision phase.

How many patients underwent imaging and at what point in their evolution?

Reviewer 3 Report

Comments and Suggestions for Authors

In their study, Daodu et al. address the pressing issue of lung injury in the acute sequelae of SARS-CoV-2 infection (PASC) or long-term COVID-19. The manuscript is well written, and the number of instrumental methods used to reflect the structural and functional state of patients' lungs is impressive. Also noteworthy is the detailed clinical characterization and well-chosen approaches to statistical data processing. However, there are some shortcomings. Suggested revisions would significantly improve the quality of the manuscript.

Suggestions:

  1. Line 20. Please define the abbreviation CT.
  2. Lines 22–23. Please remove the definitions for the abbreviations PFT, CXR, and CT; you provided them earlier.
  3. Line 32. Please define the abbreviations FEV1 and FVC.
  4. Lines 55–57. The complexity of differential diagnosis of long COVID should also be discussed in the "Introduction" section, as symptoms that develop some time after acute coronavirus infection may be associated with more than just long COVID.
  5. Lines 58–61. Pulmonary fibrosis occurs not due to excessive accumulation of extracellular matrix in the pulmonary interstitium, but due to excessive collagen deposition in the extracellular matrix of the lung parenchyma. Please correct the sentence.
  6. Lines 73–74. Please indicate in the text of the manuscript any signs of persistent inflammation on CT.
  7. Lines 188–194. Please clarify in the "Materials and Methods" section or at the beginning of this paragraph that the authors display data in Mean (Standard Deviation) format.
  8. Line 204. In Table 2, display specific p-values ​​or indicate that p<0.001. Also, in the "Percentage of Patients" column, add the "%" sign next to the numbers.
  9. Line 207. Please change "Figure 1.0" to "Figure 1."
  10. Line 213. Figure 1 contains strikethrough numbers. Please format the figure correctly.
  11. Improve the dpi quality of Figures 1, 2, 3, and 4.
  12. Lines 218–232. Please indicate in the text whether the correlations are statistically significant.
  13. Line 258. The abbreviation PASC appears twice.
  14. Line 263. The abbreviation TLCO appears twice.
  15. Lines 263–277. Why are some values ​​in bold?
  16. Please remove all repeated abbreviations in the text.
  17. Lines 325–327. Please discuss how ethnicity might be associated with the predicted lower DLCO after COVID-19.
  18. Line 338. Limitations should be added to the Materials and Methods section.

Reviewer 4 Report

Comments and Suggestions for Authors

The manuscript "A Retrospective Observational Study of Pulmonary Impairments in Long COVID Patients" by Daodu et al. is a retrospective study concerned with analyzing the pulmonary impairmement in patients with post-acute sequale of SARS-CoV 2 infection (Long Covid). The authors sought to create a set of recommendations for clincal specialists when dealing with possible long Covid patients.
This is a well-executed study, I have only some comments and suggestions on how to improve it.
1. What I missed the most in this study, is the characterization and correlation between the spirometric parameters of the patients and actual severity of Covid-19 infection that they suffered from. I understand, that it may be not possible to obtain and analyze this data in the time frame of this review (or at all), but I would appreciate if the authors at least discussed it, probable after lines 315-328, but I leave it to the judgement of the authors.

2. Description of statistical analysis, while comprehensive, is somewhat long and filled with numerical details, making it hard for comprehension. Perhaps, it would be better to reformat this part of the description (lines 143-157) into the table?

3. Figures 3 and 4 could be improved by making the text bigger and more uniform (i.e. the "normal lung" and "collapsed lung" in Figure 3 is barely legible).

Round 2

Reviewer 4 Report

Comments and Suggestions for Authors

I thank the authors for addressing my comments and concerns.